# Early Epilepsy Surgery in Benign Cerebral Tumors: Avoid Your ‘Low-Grade’ Becoming a ‘Long-Term’ Epilepsy-Associated Tumor

**DOI:** 10.3390/jcm11195892

**Published:** 2022-10-05

**Authors:** Catrin Mann, Nadine Conradi, Elisabeth Neuhaus, Jürgen Konczalla, Thomas M. Freiman, Andrea Spyrantis, Katharina Weber, Patrick Harter, Felix Rosenow, Adam Strzelczyk, Susanne Schubert-Bast

**Affiliations:** 1Epilepsy Center Frankfurt Rhine-Main, Center of Neurology and Neurosurgery, University Hospital Frankfurt, Goethe-University Frankfurt, 60528 Frankfurt am Main, Germany; 2LOEWE Center for Personalized Translational Epilepsy Research (CePTER), Goethe-University Frankfurt, 60528 Frankfurt am Main, Germany; 3Institute for Neuroradiology, University Hospital Frankfurt, Goethe University Frankfurt, 60528 Frankfurt am Main, Germany; 4Department of Neurosurgery, Center of Neurology and Neurosurgery, Goethe-University Frankfurt, 60528 Frankfurt am Main, Germany; 5Department of Neurosurgery, University Medical Center Rostock, 18057 Rostock, Germany; 6Neurological Institute (Edinger Institute), University Hospital Frankfurt, 60528 Frankfurt am Main, Germany; 7University Cancer Center (UCT), University Hospital Frankfurt, 60528 Frankfurt am Main, Germany; 8Frankfurt Cancer Institute (FCI), 60528 Frankfurt am Main, Germany; 9German Cancer Consortium (DKTK) Partner Site Frankfurt/Mainz and German Cancer Research Center (DKFZ), 69120 Heidelberg, Germany; 10Department of Child Neurology, University Hospital Frankfurt, Goethe-University Frankfurt, 60528 Frankfurt am Main, Germany

**Keywords:** seizure, ganglioglioma, dysembryoplastic neuroepithelial tumor, seizure outcome, neuropsychological outcome, cognitive deficit, cMRI

## Abstract

Epilepsy surgery in low-grade epilepsy-associated neuroepithelial tumors (LEAT) is usually evaluated in drug-resistant cases, often meaning a time delay from diagnosis to surgery. To identify factors predicting good postoperative seizure control and neuropsychological outcome, the cohort of LEAT patients treated with resective epilepsy surgery at the Epilepsy Center Frankfurt Rhine-Main, Germany between 2015 and 2020 was analyzed. Thirty-five patients (19 males (54.3%) and 16 females, aged 4 to 40 years (M = 18.1), mean follow-up 33 months) were included. Following surgery, 77.1% of patients remained seizure-free (Engel IA/ILAE 1). Hippocampus and amygdala resection was predictive for seizure freedom in temporal lobe epilepsy. In total, 65.7% of all patients showed cognitive deficits during presurgical workup, decreasing to 51.4% after surgery, predominantly due to significantly less impaired memory functions (*p* = 0.011). Patients with presurgical cognitive deficits showed a tendency toward a longer duration of epilepsy (*p* = 0.050). Focal to bilateral tonic-clonic seizures (*p* = 0.019) and young age at onset (*p* = 0.018) were associated with a higher likelihood of cognitive deficits after surgery. Therefore, we advocate early epilepsy surgery without requiring proof of drug-resistance. This refers especially to lesions associated with the non-eloquent cortex.

## 1. Introduction

The second most common histopathological finding in drug-resistant epilepsies is epilepsy-associated WHO Grade I tumors, usually with benign behavior in both adults and children [1]. Furthermore, epilepsy secondary to these tumors is often drug-resistant [2]. Following surgery for epilepsy, these tumors have a better prognosis than other histopathologic entities associated with drug-resistant epilepsies, such as hippocampal sclerosis or focal cortical dysplasia, with 74.2–89% of patients being seizure-free [3,4,5,6].

Initially, the acronym “LEAT” (long-term epilepsy-associated tumors) was proposed for this group of tumors [5,7]. By definition, “long-term” refers to drug-resistant seizures for two years or more. It has recently been shown that longer durations of epilepsy to surgery are associated with less favorable seizure outcomes in tumor-related epilepsies [8]. Thus, timely surgery for this type of epilepsy can make terms such as “chronic” or “long-term” obsolete. Accordingly, changing the acronym LEAT to “low-grade epilepsy-associated neuroepithelial tumors” has recently been proposed [9]. Due to patient diversity and small sample sizes, factors associated with seizure control remain incompletely understood. 

The histopathological spectrum of LEAT is morphologically and genetically heterogeneous. The majority of LEATs are gangliogliomas (GG) and dysembryoplastic neuroepithelial tumors (DNET), although a number of additional subtypes have been proposed [10]. Despite currently available molecular genetic markers such as BRAF V600E and FGFR1 and immunohistochemical surrogate markers such as CD34 and p16, there is still a poor inter-rater agreement in the histopathological diagnosis [11]. On the other hand, the clinical characteristics of histopathologically distinct tumor entities are still incompletely understood.

Gross-total resection, including a rim of normal-appearing tissue, has been demonstrated to be superior to lesionectomy alone in postoperative seizure control [8]. However, the extent of wide resection is poorly defined. Many centers utilize intraoperative electrocorticography (ECoG) to define resection margins, including the complete irritative zone if possible. However, the exact utility of EcoG is controversial. In tumors located in the temporal lobe, the resection of mesiotemporal structures appears to lead to improved seizure control, but at the cost of greater cognitive deficits [12,13]. Predictive factors for good seizure control and risk factors for postoperative functional and cognitive deterioration should be identified as exactly as possible for counseling and selecting patients who will benefit from early surgery. 

This retrospective, monocenter study aimed to identify the distinct neuroradiologic, histopathologic, and clinical profiles of LEAT and their association with postsurgical seizure and neuropsychological outcomes. We analyzed the impact of tumor size and location on cognitive and seizure outcome. Furthermore, another aim of this study was to evaluate the use of intraoperative ECoG in determining resection margins. 

## 2. Patients and Methods

All patients diagnosed with LEAT in accordance with the WHO histopathological criteria [9] treated with resective epilepsy surgery in the Epilepsy Center Frankfurt Rhine-Main, Germany were included in this study. Surgeries were performed between November 2015 and November 2020. All epilepsy surgeries were continuously recorded and evaluated to meet the quality specifications of the Austrian, German, and Swiss working group on presurgical epilepsy diagnosis and operative epilepsy treatment [14]. Patients were identified from this database, and a retrospective electronic chart review was performed to ensure that all LEAT cases were included. Information was obtained from presurgical and postsurgical medical records and medical letters. 

Preoperative clinical evaluation was performed according to a standardized protocol comprising medical history, seizure semiology, neurological examination, and neuropsychological assessment. Long-term video-EEG monitoring was performed in all patients to record typical seizures and evaluate semiology, ictal onset zone, and interictal epileptiform activity. In two patients, presurgical invasive EEG monitoring using sEEG electrodes was performed. The localization and extent of the underlying epileptogenic lesion were evaluated by 1.5 or 3 Tesla MRI. Language lateralization was examined by functional transcranial Doppler sonography [15]. If necessary, functional MRI or Wada testing was also carried out. Based on these findings, surgery was individually indicated and planned in our interdisciplinary epilepsy surgery conference. Patients were classified into either “drug-resistant” or “non-drug resistant”, following the definition of drug-refractoriness laid down by the ILAE [16]. 

The operative procedures aimed to remove both the epileptogenic zone and the lesion. Resection types included ECoG-guided topectomy (24 of 35 patient cases), classical 2/3-temporal lobe resection (seven cases), and temporal pole resection (four cases). 

Intraoperative ECoG was evaluated by two of four certified neurophysiologists (CM, FR, AS, and SSB). During a light and steady level of gas anesthesia, a strip electrode was placed on the cortex. Registration was performed over several minutes until the ECoG showed a continuous pattern (i.e., no burst suppression). ECoG was recorded during a pre-resection phase and after the tumor resection phase and epileptogenic zone resection. ECoG findings were categorized as “no spiking” or “with spiking” for each recording period and electrode. Intraoperative motor and somatosensory-evoked potentials were used to monitor the integrity of motor and somatosensory function if required. 

Standardized assessments of seizure frequency and neurological and neuropsychological functions were performed before surgery (baseline) and at six months, one year, and two years after surgery (i.e., routine follow-up). Seizure outcomes were evaluated using the Engel and International League Against Epilepsy (ILAE) classifications [17]. Histopathological reviews of resected brain tissues were performed at the local Department of Neuropathology (Edinger Institute), and the diagnoses in two patients were independently confirmed by two German reference centers. Neuropsychological test bat-teries assessing the relevant cognitive domains (i.e., attentional functions, verbal learning and memory, nonverbal learning and memory, and executive functions) were performed in both children and adult patients [18,19], and relevant cognitive deficits were defined as *z*-value ≤ 1.0. Additionally, two self-report questionnaires for assessing symptoms of depression (Beck Depression Inventory-II [BDI-II]) [20] and quality of life (Quality of Life in Epilepsy Inventory-31 [QOLIE-31]) [21] were performed where applicable. We have previously described the details of our presurgical evaluation protocol, including video-EEG [22].

The study was approved by the Ethics Committee of the Goethe-University Frankfurt (reference 20-951_1). Informed consent was waived due to the retrospective nature of the study.

### 2.1. Neuroimaging

Preprocessing and tumor segmentation were performed on preoperative MR images (T1w, CE-T1w, T2w, and FLAIR) using the BraTS toolkit [23]. All segmentations were reviewed by a neuroradiologist (EN) and manually corrected if necessary. Further analysis was performed using the FMRIB Software Library 6.0 [24]. The three-dimensional segmentation masks were used to calculate the tumor volume of each patient. Furthermore, a tumor localization map of all affected voxels across patients was calculated based on the individual segmentation masks. For this purpose, the T1w image of each patient was brought into the standard Montreal Neurological Institute (MNI) space using a nonlinear registration, and the resulting nonlinear warp fields were applied to each segmentation mask. Finally, all segmentation masks in the MNI space were summed. In addition, patients were divided into two groups according to their postoperative seizure outcome (Engel IA vs. >Engel IA), and the respective segmentation masks were summed and visualized in different colors.

A neuroradiologist (EN) and an epileptologist (CM) visually examined the postoperative MR images for the extent of resection. Additionally, they recorded whether the hippocampus, amygdala, and parahippocampal gyrus were also resected.

### 2.2. Statistical Analysis

Several neuroradiologic, histopathologic, and clinical factors were analyzed with regard to their association with postsurgical seizure and neuropsychological outcomes, using chi-square and Mann–Whitney U tests. Cochran’s Q, and paired-samples Wilcoxon, tests were used to detect differences between the neuropsychological assessment outcomes (cognitive deficits, depression symptoms, and quality of life) before and after surgery.

## 3. Results

Thirty-five patients were included in this study, with age at surgery ranging from four to 40 years (*M* = 18.1, *Mdn* = 17, *SD* = 8.1); eighteen patients (51.4%) were younger than 18 years at the time of surgery. Sixteen patients (45.7%) were female, and 19 (54.3%) were male. A summary of the clinical characteristics of our cohort is provided in Table 1. All patients presented with seizures, which were the only symptom of their tumor. The mean age at epilepsy manifestation based on patient or caregiver self-report was 11.5 years (*Mdn* = 14, *SD* = 5.5, range: 1–22), and the mean epilepsy duration at the time of surgery was six years (*Mdn* = 4, *SD* = 5.5, range: 0.25–25). On average, patients failed to respond to 3.8 antiseizure medications (ASM) before presurgical diagnosis (*Mdn* = 4 ASD, *SD* = 1.8 ASD, range: 1–8 ASD).

The epileptogenic lesion was located in the temporal lobe in 27 (77.1%) patients, and eight (22.9%) patients suffered from extratemporal epilepsy (*n* = 5 with frontal lobe epilepsy, *n* = 3 with occipital lobe epilepsy). Four patients underwent presurgical evaluation and subsequent epilepsy surgery before fulfilling the criteria for drug-resistant epilepsy. The epilepsy surgery performed at our center constituted a reoperation in five cases, following earlier resective brain surgery in 4/5 patients performed at external neurosurgery departments, predominantly without prior pre-surgical work-up. 

Ganglioglioma was the most common histopathological diagnosis, diagnosed in 19 patients, including two with a dual histopathologic diagnosis (ganglioglioma plus hippocampal sclerosis and ganglioglioma as part of FCD Type IIIb). In 13 patient cases, a DNET was diagnosed histopathologically. One patient in case each was identified as having an angiocentric glioma WHO Grade I, one as having an oligodendroglioma WHO Grade II, and one having a low-grade glial tumor with desmoplastic reaction. Gangliogliomas were located in the temporal lobe significantly more frequently than DNETs (94.7% vs. 61.5%, *p* = 0.018).

The localization of all LEATs projected onto a standard brain is shown in Figure 1A. Mean tumor volume was 14.04 cm^3^ (*SD* = 19.19, *Mdn* = 6.53, range: 0.16–89.01), with no statistically significant differences between histopathologic diagnoses. 

During resection of temporally located LEATs, the mesiotemporal structures (including the amygdala and hippocampus) were completely removed in 55.5% of patients. In one-third of patients, the mesiotemporal structures were fully preserved under surgery, and in 11.1% of the patients, a partial resection was performed. In ten (37.0%) patients, the parahippocampal gyrus was either preserved or completely resected, and partial resection was performed in 26% of cases. 

ECoG was used in 68.6% (*n* = 24) of operations to determine resection margins. In 17 (48.6%) cases, complete original ECoG data were available, with ECoG recordings recorded pre and post resection. Only patients with complete ECoG data pre and post resection were included in the statistical analysis. In each ECoG recorded before resection, spiking could be detected over the tumor area. In 47 % of cases (*n* = 8), the ECoG results showed persistent spiking after the initial resection, so a multidisciplinary intraoperative decision was made to extend the resection margins further. After the final resection, ECoG did not record any spiking in 82.4% (*n* = 14) of cases, whereas spikes could still be recorded on the brain surface in 17.6% (*n* = 3) of cases. Further resections were not performed in these cases, primarily due to the proximity to the eloquent cortex. According to the first postoperative MRI, 27 of the 35 LEATs (77.1%) had been completely resected. Figure 2 demonstrates examples of ECoG recording performed during epilepsy surgery for LEAT. 

### 3.1. Seizure Outcome

The mean follow-up was 33 months. Outcome data were available at six months for all patients, at one year for 97% of patients, and at two years or more for 80% of patients. At the most recent visit, 27 patients remained completely seizure-free (77.1% Engel IA/ILAE 1), and 91.4% were free of disabling seizures (Engel I). In 17 (48.6%) patients, ASMs could be completely discontinued. Figure 1B shows the localization of LEATs in relation to seizure outcome (Engel IA vs. >Engel IA).

According to chi-square tests, patients without spiking in the ECoG following the final resection showed a tendency toward better seizure outcomes after surgery compared to patients with remaining spikes in the ECoG (85.7% vs. 33.3% Engel IA, *p* = 0.052). Complete resection of the hippocampus and amygdala in temporal lobe epilepsy was associated with a favorable outcome (*p* = 0.047). There was no significant association between postoperative seizure freedom and the occurrence of focal to bilateral tonic-clonic seizures (FBTCS) before surgery (*p* = 0.865), the occurrence of sharp waves in the EEG six months after surgery (*p* = 0.370), the resection of the parahippocampal gyrus (*p* = 0.479), extratemporal tumor location (*p* = 0.382), or the completeness of LEAT resection (*p* = 0.687). Mann–Whitney U tests revealed no significant association between postoperative seizure outcome and duration of epilepsy (*p* = 0.558), the number of ASMs before surgery (*p* = 0.614), or tumor size (*p* = 0.281).

### 3.2. Neuropsychological Outcome

Before surgery, relevant deficits (i.e., *z*-value ≤ –1.0) in one or more of the assessed cognitive domains were found in 65.7% (*n* = 23) of all patients. There were impairments in attentional functions (22.9% of all patients), verbal learning and memory (14.3%), nonverbal learning and memory (45.7%), and executive functions (25.7%) before surgery. Patients with, compared to those without, cognitive deficits before surgery showed a tendency toward a longer duration of epilepsy (*M* = 7.4 vs. 4.1 years, *p* = 0.050). 

It was possible to obtain a complete neuropsychological follow-up for 28 patients (80.0%). The mean follow-up was 21 months (*SD* = 5.3). The proportion of all patients showing relevant deficits in one or more of the assessed domains decreased to 51.4% (*n* = 18) after surgery (Table 2). The cognitive improvement in patients was found beginning at the 6-month follow-up with maximum improvement at the 12-month follow-up, after which the cognitive functions remained stable. The Cochran’s Q tests revealed a significant decrease in relevant deficits in nonverbal learning and memory (*p* = 0.011) after surgery.

Paired-samples Wilcoxon tests revealed a significant decrease in symptoms of depression assessed by the BDI-II (*Mdn* score = 7.0 vs. 2.5, *p* = 0.033), and a significant increase in quality of life assessed by the QOLIE-31 (*Mdn* score = 58.0 vs. 59.0, *p* = 0.028) after surgery.

In the subgroup of patients with temporal lobe epilepsy (*n* = 27), chi-square tests revealed that those with FBTCS before surgery showed temporal cognitive deficits (i.e., impairments in verbal or nonverbal learning and memory) one year after surgery significantly more often than patients without FBTCS (66.7% vs. 18.2%, *p* = 0.019). According to Mann–Whitney U tests, temporal lobe epilepsy patients with postsurgical temporal cognitive deficits were significantly younger at disease onset than patients without deficits (*Mdn* = 13.5 vs. 16.0 years, *p* = 0.018). No significant associations between postsurgical temporal cognitive deficits and epilepsy duration (*p* = 0.483), the resection of mesiotemporal structures (*p* = 0.099), tumor size (*p* = 0.166), or postsurgical discontinuation of ASM (*p* = 0.078) were found.

## 4. Discussion

This study’s patients with epilepsy due to LEAT showed good postoperative seizure control, with 77.1% achieving the outcome Engel IA/ILAE 1, and 91.4% were free of disabling seizures (Engel I). This finding is in line with other studies reporting 72–77.5% of patients being seizure-free at two years follow-up [3,25]. Complete resection of the tumor does not always result in improved seizure control compared to incomplete resection; however, complete resection of the epileptogenic zone appears crucial. Intraoperative ECoG might help detect the margins of the epileptogenic zone, as in our study patients without spiking in the ECoG following the last resection had a tendency toward improved seizure outcome after surgery in comparison to patients with spikes in the ECoG after resection.

### 4.1. Predictors of Seizure Freedom

Several factors influencing seizure outcome have been described previously. Gross-total resection is predictive of complete seizure freedom when compared with subtotal resection [26,27]. Other predictors of seizure freedom include the duration of epilepsy, age at surgery, and preoperative seizure control on ASM [2,6]. Our cohort of patients showed a short duration of epilepsy to surgery compared to prior studies, with a median epilepsy duration of four years. Therefore, we could not detect a statistically significant effect of duration on seizure outcome. Additionally, likely due to the short average latency to surgery, our cohort showed a relatively good postoperative outcome with 91.4% Engel I compared to 74.2% to 89.7% [2,4,5,6,28]. The long-term analysis of year-to-year seizure outcome demonstrates a highly stable percentage over an 11-year follow-up; malignant transformation of a LEAT is rare and appears more frequent in GG [5]. Extratemporal tumor manifestation and incomplete resection have been described as negative factors regarding postoperative seizure control [5,6]. Postoperative, seizure-free outcomes significantly decrease in tumor locations involving eloquent areas, with incomplete tumor resection as the leading cause of surgical failure [29]. Although we also observed this trend in our cohort, it was not statistically significant due to small sample sizes (Figure 1B).

The ECoG is often considered helpful in intraoperatively defining the resection margins of a gross-total resection. After an initial period of skepticism, there has been renewed interest in ECoG for intraoperatively guiding epileptogenic tissue resection to increase the odds of a favorable postoperative seizure outcome. In temporal and extratemporal lesional epilepsies, especially when relating to focal cortical dysplasia, tuberous sclerosis, or cavernous malformations, there was found to be a strong correlation between ECoG-guided resection and postoperative seizure relief [30]. The study evidence for LEATs appears to be heterogeneous. A retrospective study evaluating the use of ECoG during tumor resection in 119 pediatric patients suggested that ECoG does not provide improved seizure freedom in children compared to lesionectomy alone. However, preoperative seizure duration and the number and duration of ASMs were significantly higher in the ECoG-guided resection group than in the group without ECoG [31]. Others reported ECoG-guided tailored epilepsy surgery to be superior to lesionectomy alone in LEAT patients [32,33]. However, whether ECoG was preferentially utilized in cases involving more severe epilepsy or for lesions less amenable to gross-total resection—actors potentially confounding this result—could not be discerned given the lack of data disaggregation in the literature. In our cohort, ECoG was utilized in the majority of patients. Spiking in the ECoG after the final resection was associated with a less favorable seizure outcome compared to patients with no remaining spikes during the last ECoG. This result indicates that ECoG could be a useful tool in detecting the extension of the epileptogenic zone and, in some cases, may also help preserve hippocampal structures.

At the histopathological level, our knowledge of the molecular-genetic and immunohistochemical profiles of LEATs has increased in recent years. An improved outcome in CD34-positive tumors has been described, with GG and pleomorphic xanthoastrocytoma being the histologic types with the strongest association with CD34-positivity compared with DNETs [34]. The oncogenic BRAF V600E somatic mutation frequently found in epilepsy-associated pediatric brain tumors arises during early brain development and causes intrinsic epileptogenicity in developing neurons [35]. However, the clinical correlations and consequences of molecular genetic distinct tumor subtypes remain poorly understood, so future research into personalized treatment regimens is required. Due to its rarity, the incidence of each LEAT subgroup is quite limited in single centers; therefore, prospective multicenter studies are needed to understand and analyze these associations further.

### 4.2. Cognitive Decline in Patients with LEAT and Improvement following Epilepsy Surgery

Before surgery, 65.7% of all patients in our cohort showed relevant deficits in one or more of the assessed cognitive domains (i.e., attentional functions, verbal learning and memory, nonverbal learning and memory, and executive functions). Patients with cognitive deficits before surgery showed a tendency toward a longer duration of epilepsy than those without cognitive deficits. The proportion of all patients showing relevant cognitive deficits decreased to 51.4% after surgery.

FBTCS were associated with a higher likelihood of cognitive deficits, presumably due to the more pronounced network dysfunction. Likewise, an association between a young age of disease onset and cognitive deficits was found, which may indicate increased susceptibility to epileptic activity in the developing brain. Older age at seizure onset has been previously found to be associated with higher cognitive functioning [28]. No significant difference by tumor volume was shown. 

In line with previous studies, there was a significant decrease in symptoms of depression, and a significant increase in quality of life in our cohort after surgery [36].

A similar proportion of patients with LEATs in the temporal lobe have been frequently reported to show cognitive impairments before surgery, with impairments in preoperative memory in 67.1%, executive function in 44.7%, and language in 45.5% [37]. Predictors for pre- and postoperative cognition in this study mainly corresponded with what is already known about temporal lobe epilepsy and resections in general, with higher baseline performance; left side, mesial location; and hippocampal resection relating to postsurgical memory decline. In a pediatric case series, cognitive functioning deteriorated with time in glioneuronal tumor-related refractory epilepsy [38]. Early epilepsy onset, irrespective of epilepsy duration, is related to circumscribed cognitive dysfunction in verbal but not nonverbal memory. In contrast, longer epilepsy duration, irrespective of age at epilepsy onset, is related to presurgical lower overall cognitive functioning in children [39]. Therefore, timely epilepsy surgery is vital to preserve a child’s brain’s compensatory and developmental capacities as much as possible. However, it remains unclear whether different LEAT types are associated with different cognitive and seizure outcomes as our cohort was too small to detect subtle differences.

Previously, epilepsy surgery was reserved for treatment-refractory patients. DNETs and gangliogliomas are highly epileptogenic, and the frequency of seizures reaches to almost 100% in those with DNETs and to 80–90% in those with GGs [40]. The proportion of drug-resistant patients is high. However, one challenge in clinical practice is the fact that “treatment refractoriness” is a function of time. This course of disease may be divergent between patients, with an earlier onset of drug-refractoriness in some patients and later in others. However, cognitive functions are impaired in many patients even without a refractory course of epilepsy. Surgery should not be withheld from this patient group, in order to preserve or even improve cognitive functions.

### 4.3. Limitations of This Study

This was a retrospective mono-center study. Subgroups of individual LEATs and localizations were very small, making subgroup analysis challenging to impossible. Due to the rarity of the disease, multicenter studies are needed to more accurately measure the effects of individual factors. Due to the limited number of patients, the cohort investigated in this study still remains histopathologically heterogeneous. However, it may be the basis for larger meta-analyses or multicenter studies. Further genetic and histoarchitectonic differences between tumor entities can also be studied and analyzed to better understand these distinct cases in the future. 

In the study cohort, one child with histopathologically confirmed oligodendroglioma WHO grade II was included, which does not necessarily fulfill the LEAT criteria in the strictest sense. However, since WHO grade II tumors in children belong to the group of benign brain tumors with good prognosis, the treatment of this patient group is not different from the treatment of patients with LEAT in the strictest sense, so we decided to include this patient. 

## 5. Conclusions

Our results add support to the currently available data showing that LEATs have an overall good postoperative prognosis. Despite the short course of the disease on average, a high proportion of patients show cognitive deficits. In order to avoid jeopardizing the favorable postoperative prognosis and further cognitive decline, we advocate early epilepsy surgery in appropriate patients. Suitable patients, who may not need to wait for evidence of having a drug-resistant course, are those with LEATs in non-eloquent areas, primarily temporal. ECoG could be a useful tool in defining resection margins. Further studies, preferably multicenter, are needed to better understand prognostic factors and offer early intervention to appropriate patients.

## Figures and Tables

**Figure 1 jcm-11-05892-f001:**
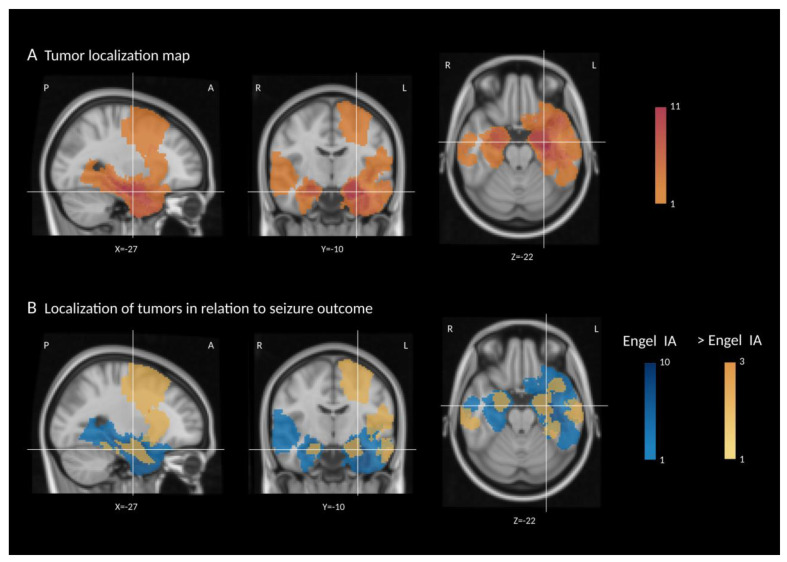
(**A**) Tumor localization map—summed segmentation masks of all patients overlaid on the standard brain image (MNI). The legend refers to the number of patients with tumor tissue in a voxel, with darker red indicating a higher number of patients. (**B**) Localization of tumors in relation to seizure outcome. Patients were divided into two groups according to their postoperative seizure outcome, and the respective segmentation masks were summed. Legends refer to the number of patients with tumor tissue in a voxel, with darker blue indicating a higher number of patients with outcome Engel IA and darker orange a higher number of patients with outcome >Engel IA. X, Y, and Z represent MNI space coordinates for the given slice. L = left, R = right, A = anterior, and P = posterior.

**Figure 2 jcm-11-05892-f002:**
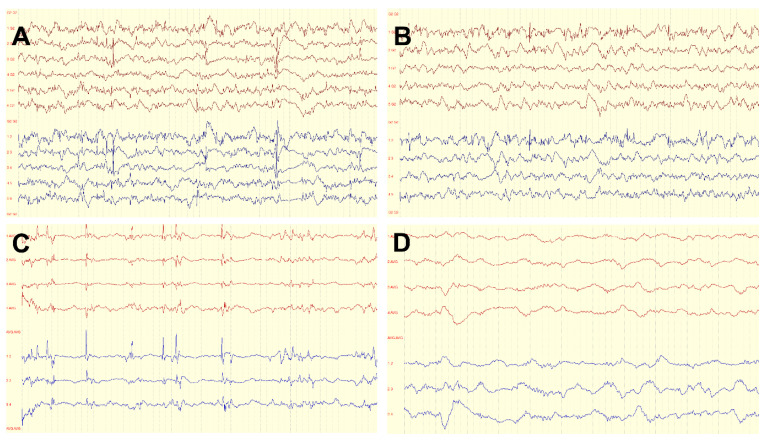
Intraoperative ECoG findings in patients with epilepsy due to LEAT. Red lines: monopolar montage, blue lines: bipolar montage. (**A**) A 16-year-old male with epilepsy since six months, presenting with first bilateral tonic-clonic seizures and suspected DNET in the right frontal lobe. ECoG before resection showing frequent spiking with maximum at Electrode 2. (**B**) The same patient following tumor resection; ECoG showing persistent spiking with maximum at Electrode 1. Gross total resection was limited in the frontotemporal resection margins due to a suspected dominant right hemisphere; patient had a seizure relapse one-year post surgery following drug withdrawal. (**C**) A six-year-old girl suffering from left temporal lobe epilepsy since two years of age; histopathology confirmed ganglioglioma with associated focal cortical dysplasia. ECoG before resection with frequent spiking. (**D**) The same patient after gross-total resection showing diffuse slowing without spiking; patient is seizure-free, and ASMs are reduced.

**Table 1 jcm-11-05892-t001:** Clinical characteristics and outcome of the patient cohort (*n* = 35).

Patient No	Sex	Age	Age at Onset	Epilepsy Duration	Side Left = 1 Right = 0	Temporal Lobe Epilepsy 1 = Yes 0 = No	FBTCS before Surgery 1 = Yes 0 = No	Histopathology GG = 1 DNET = 2 Other = 3	No of ASM, Total	No of ASM, at Time of Surgery	ASM after Surgery None = 1, Reduced = 2 Unchanged = 0	Complete Resection According to Post-Op MRI 1 = Yes 0 = No	SW in 6-Months Follow Up EEG 1 = Yes 0 = No	Engel/ILAE Most Recent Visit	Follow Up (Months)	Tumor Volume (mm^3^)
1	m	4	3	1	0	1	0	2	4	2	2	0	1	IA/1	48	38,934
2	f	6	2	4	1	1	0	1	2	2	1	1	0	IA/1	24	6532
3	m	6	5	1	0	1	0	2	1	1	1	0	0	IA/1	24	89,092
4	f	7	5	2	1	0	1	2	2	2	1	1	0	IA/1	40	3842
5	f	8	8	1	1	0	1	3	1	1	1	1	0	IA/1	18	2852
6	m	9	8	2	1	1	0	2	4	2	2	1	0	IA/1	20	18,484
7	f	12	11	2	1	0	1	2	3	2	1	0	0	IA/1	48	5003
8	m	12	13	0	0	1	1	2	2	2	1	1	0	IA/1	12	20,936
9	f	12	7	5	1	0	0	1	5	2	0	0	1	IIIA/4	12	53,120
10	f	13	3	11	0	0	0	2	3	2	1	1	1	IA/1	36	7909
11	m	14	8	6	1	1	0	1	3	2	2	1	0	IA/1	12	1494
12	m	15	15	1	1	1	0	2	1	1	1	1	0	IA/1	24	46,627
13	m	16	14	3	1	1	1	1	6	2	1	1	0	IA/1	24	4367
14	m	16	15	0	0	0	1	2	1	1	0	1	0	ID/3	24	1336
15	f	16	14	2	1	1	1	1	3	2	1	1	0	IA/1	30	6991
16	f	16	14	2	1	0	1	3	3	1	1	1	1	ID/3	18	49,102
17	m	17	14	3	0	1	0	1	2	2	1	1	0	IA/1	60	8196
18	m	17	1	16	1	1	1	1	6	1	0	0	0	IB/2	6	9135
19	m	18	15	3	1	1	1	1	6	2	0	1	1	IIA/3	48	4259
20	m	18	14	3	1	1	0	1	3	1	0	1	0	IB/2	48	159
21	m	19	11	8	1	1	1	1	5	2	1	0	0	IA/1	60	6836
22	f	19	7	12	0	0	1	2	5	2	2	0	0	IA/1	36	8875
23	m	20	14	7	1	1	0	1	4	2	1	1	1	IA/1	48	21,044
24	f	20	13	7	0	1	1	1	5	3	2	1	0	IA/1	30	4061
25	f	22	17	6	1	1	0	1	4	2	1	0	0	IA/1	42	3404
26	f	22	17	5	1	1	1	2	7	2	1	1	0	IA/1	60	10,893
27	m	23	19	4	0	1	0	1	5	2	2	1	0	IA/1	36	923
28	f	24	17	7	0	1	0	1	4	1	0	1	0	IIA/3	36	3099
29	f	25	2	2	0	1	1	3	8	3	2	1	0	IA/1	60	1228
30	m	26	9	17	1	1	1	1	3	1	1	1	0	IB/2	24	3859
31	m	28	16	12	1	1	0	1	3	2	0	1	0	IA/1	24	29,233
32	f	30	22	8	1	1	1	1	4	2	2	1	0	IA/1	24	1054
33	m	31	15	10	1	1	1	1	4	2	2	1	0	IA/1	36	11,003
34	f	33	21	13	0	1	0	2	3	2	2	1	0	IA/1	36	2753
35	m	40	15	25	1	1	1	2	8	2	0	1	0	IA/1	12	4635
		*Mdn* = 17	Md = 14	*Mdn* = 4	left: 23	total *n* = 27	total *n* = 19		*Mdn* = 4	*Mdn* = 2	unchanged = 8	total *n* = 27	total *n* = 6	*n* = 27 IA/1	*Mdn* = 30	*Mdn* = 6532

Abbreviations: 1: yes, 0: no, m: male, f: female, FBTCS: focal to bilateral tonic clonic seizure, GG: gangioglioma, DNET: dysembryoplastic neuroepithelial tumor, ASM: antiseizure medication, and *Mdn*: Median.

**Table 2 jcm-11-05892-t002:** Proportion of patients showing deficits in each assessed cognitive domain, compared between assessments using Cochran’s Q tests.

Cognitive Domains Assessed	Before Surgery	After Surgery	*p*-Value
Attentional functions (*n* = 21)	22.9%	14.3%	0.102
Verbal memory (*n* = 23)	14.3%	17.1%	0.414
Nonverbal memory (*n* = 23)	45.7%	17.1%	0.011 *
Executive functions (*n* = 23)	25.7%	20.0%	0.999
OVERALL relevant deficits in one or more domains (*n* = 28)	65.7%	51.4%	0.564

* *p* < 0.05.

## Data Availability

The data presented in this study are available on request from the corresponding author.

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
