# Peer review of "Early Epilepsy Surgery in Benign Cerebral Tumors: Avoid Your ‘Low-Grade’ Becoming a ‘Long-Term’ Epilepsy-Associated Tumor"

_jcm, 2022, doi:10.3390/jcm11195892_

Round 1
Reviewer 1 Report
- only few typos, spell check required / e.g. inconsistent use of capital letters in specific terms (as Low grade epilepsy associated tumors)
- last sentence of abstract: not the lesions themself are non-eloquent but the cortex in which the lesions are located. Please change.
- Please consider adding a patient characteristics table within the main text.
- Please add the number of patients that had been operated on without proof of pharmacoresistancy
- it would be interesting to add the number of patients that had been evaluated for epilepsy surgery at your center within the time period but did not proceed to surgery
- p2 l59: Please change the wording "worse seizure free outcome"
- 4.2 cognitive decline.... line 346: how is "relevant" cognitive deficits defined?
- Limitations: Please discuss the limitations by a heterogeneous cohort (age, type and location of tumor) in more detail. Within introduction you state (line 53) "due to patient diversity and small sample sizes, factors associated with seizure control remain incompletely understood." This issue is not completely solved by your study and should therefore be mentioned in limitations.
Reviewer 2 Report
I thought the contents of the paper justified your case for earlier surgery to remove epileptogenic tissue in relation to ASM treatment-refractory epileptic seizures in persons with low grade neuroepithelial tumours. However, I wondered if your data would permit determination of when the post-surgery improvement in cognition occurred. If early, the benefit might arise from tumor tissue or seizure cessation, but if later, only after ASM dosage reduction or cessation, the interference would probably be medication-related, and a further reason for earlier surgery.
However, with earlier operation, you would have probably no longer have the ASM treatment-failure as an indication for operating. Do you have data on the proportion of these neuroepithelial tumours that produce seizures that are controlled by ASM therapy? Your earlier surgery idea might come to involve such persons.
